# Construction of Ecological Security Pattern Based on the Importance of Ecological Protection—A Case Study of Guangxi, a Karst Region in China

**DOI:** 10.3390/ijerph19095699

**Published:** 2022-05-07

**Authors:** Yanping Yang, Jianjun Chen, Renjie Huang, Zihao Feng, Guoqing Zhou, Haotian You, Xiaowen Han

**Affiliations:** 1College of Geomatics and Geoinformation, Guilin University of Technology, Guilin 541004, China; yangyp@glut.edu.cn (Y.Y.); huangrj@glut.edu.cn (R.H.); fengzihao@glut.edu.cn (Z.F.); gzhou@glut.edu.cn (G.Z.); youht@glut.edu.cn (H.Y.); xwhan@glut.edu.cn (X.H.); 2Guangxi Key Laboratory of Spatial Information and Geomatics, Guilin University of Technology, Guilin 541004, China

**Keywords:** ecosystem service function importance assessment, ecological sensitivity assessment, circuit theory, ecological corridor, ecological security pattern

## Abstract

The ecological security pattern is an important way to coordinate the contradiction between regional economic development and ecological protection and is conducive to promoting regional sustainable development. This study examines Guangxi, a karst region in China. The ecosystem service function and ecological environment sensitivity were both selected to evaluate the ecological conservation importance, and based on the results of the ecological conservation importance evaluation, suitable patches were selected as ecological sources. Meanwhile, resistance factors were selected from both natural factors and human activities to construct a comprehensive resistance surface, circuit theory was used to identify ecological corridors, ecological pinch points, and ecological barrier points, and ecological protection suggestions were then proposed. The results show that there are 50 patches of ecological sources in Guangxi, with a total area of 60,556.99 km^2^; 115 ecological corridors, with the longest corridor reaching 194.97 km; 301 ecological pinch points, whose spatial distribution is fragmented; and 286 ecological barrier points, most of which are concentrated in the central part of Guangxi. The results of this study provide a reference for the construction of ecological security patterns and ecological conservation in developing countries and karst areas.

## 1. Introduction

In recent years, due to rapid socio-economic development and population growth, the scale of cities has been expanding, and a large amount of ecological land has been transformed into economically valuable construction land [1]. Rapid urbanization has led to economic prosperity, but it has also triggered a series of ecological and environmental problems [2,3], such as extreme climate events [4], landscape fragmentation [5], biodiversity loss [6], ecosystem degradation [7], etc. The resulting ecological problems not only affect the sustainable development of the region but also seriously threaten the safety of human beings themselves [8]. Therefore, safeguarding and maintaining ecological security and building an ecological security pattern are of great significance for correctly understanding ecological security and formulating appropriate ecological environmental protection policies.

Ecological security research has developed from ecological risk analysis. International research on ecological security has mainly focused on ecological safety evaluation [9], ecological security pattern construction [10], and ecological safety planning and design [11]. Among them, the construction of ecological security patterns is currently a research hotspot and a focus in the field of ecological security research; it is also one of the effective ways to deal with rapid urbanization and improve the regional ecological security situation [12]. An ecological security pattern is a potential spatial pattern of ecosystems in a landscape that supports both territorial development and ecological protection. International research around ecological security patterns has mainly focused on the establishment of protection systems and the identification of protective measures by developing tiered levels of protection [13]. The early construction of ecological security patterns was mainly aimed at protecting biodiversity [14]. With the deepening of people’s understanding of ecological security patterns, the study gradually changed to a focus on ecosystems [15], concentrating on the coupled relationship between ecological processes and functions. Ecological security pattern research by Chinese scholars mainly centers on the field of pattern identification and construction but has also focused on topics such as the delineation of ecological functional areas in conjunction with the actual situation in China [16].

The construction of ecological security patterns helps to coordinate the contradiction between regional socio-economic development and ecological protection by focusing on the protection of ecological sources; it also helps to permit economic construction to the maximum extent while protecting ecological sources. At present, ecological security pattern construction methods have been increasingly improved, and research based on “source identification-resistance surface construction-corridor extraction” is the basic paradigm and research framework of ecological security pattern construction [17]. Ecological sources are important ecological patches that promote ecological processes and maintain regional ecological security. Based on different research objectives and research needs, the identification methods of ecological sources are qualitative evaluation based on ecosystem structure or quantitative evaluation with comprehensive criteria. The former directly treats important ecological sites such as nature reserves [18], forest parks [19], scenic areas [20], etc., as ecological source sites based on the ecological status of the study area. Although this qualitative evaluation saves costs to a large extent, it ignores the dynamic changes within ecological land. In order to improve this problem, scholars have proposed the use of comprehensive evaluation indicators to quantitatively identify ecological sources, including ecosystem service functions [21], ecological sensitivity [22], landscape connectivity [23], and other indicators. The construction of the resistance surfaces is a prerequisite for ecological corridor extraction, and it is mainly based on the single-assignment method of land use [24] and the integrated multi-indicator assignment method [25]. Among them, the single-assignment method of land use lacks spatial heterogeneity and cannot quantify the influence of human activities on the construction of resistance surfaces, while the multi-source indicator assignment method can fully reflect the distribution of regional ecological resistance by considering the influence of both natural factors and human activities on resistance surfaces. Ecological corridors refer to the components of the ecosystem that are distributed in strips or lines in the ecological environment, can connect relatively isolated ecological patches, and can meet the energy flow and exchange between species [26]. The extraction of corridors mostly utilizes methods such as the minimum cumulative resistance model [27], graph theory method [28], and circuit theory [29]. Among them, the minimum cumulative resistance model (MCR) simulates the minimum cumulative resistance pathway by calculating the cost overcome by the species from the ecological source to the destination, thereby constructing an ecological network [30]. The graph theory method uses a series of nodes and lines to reflect the organic connections of the landscape, forming a complex ecological network [31]. In contrast, circuit theory simulates the migration process of species between ecological sources based on the wandering characteristics of current species in the circuit, and identifies ecological pinch points and ecological barrier points based on cumulative current values and cumulative current recovery values. The random wandering nature of circuit theory is more consistent with the behavioral characteristics of species, so circuit theory has become a popular method for constructing ecological corridors [32].

At present, most studies on ecological security patterns focus on areas with developed economies and intense human activities, while relatively little research has been conducted on karst areas with backward economies and more fragile ecological environments. For developing countries, economic development will inevitably lead to the deterioration of the ecological environment, and how to reconcile economic development and ecological protection is an urgent problem to be solved. Guangxi is mountainous, hilly, and intricate with unique karst landscape. It is one of the ecologically fragile regions in western China which has serious rocky desertification problems, poor soil, and is prone to soil erosion [33]. At the same time, the economy of this area is relatively backward; tourism is an essential economic source for Guangxi. Therefore, both the protection of ecological environment safety as well as economic development in Guangxi must be properly coordinated and developed together. The construction of Guangxi’s ecological security pattern is conducive to the sustainable development of the region and provides implementable decision-making suggestions for the optimal layout of the region’s territorial space.

This study combines the actual situation of the ecological environment in Guangxi; identifies ecological sources based on the importance of ecosystem service function and ecological environment sensitivity; selects appropriate natural factors and socio-economic factors to construct a comprehensive resistance surface; and adopts circuit theory to identify ecological corridors, ecological pinch points, and ecological barrier points so as to construct the ecological security pattern of Guangxi and propose suitable protection advice. Overall, this study provides a new research framework and reference for constructing ecological security patterns in economically backward and ecologically fragile regions.

The structure of the remaining part of the manuscript begins with an overview of the study area, data sources, and pre-processing. Then, the research methods are introduced, including evaluation of the importance of ecosystem service functions, ecological environment sensitivity assessment, evaluation of the importance of ecological protection, ecological source identification and resistance surface construction, ecological corridor construction, and identification of pinch points and barrier points. Thirdly, the research results are introduced, including evaluation of the importance of ecological protection and the construction of ecological security patterns. Finally, the above results are discussed, and the deficiencies and future improvement measures are pointed out.

## 2. Materials and Methods

### 2.1. Study Area

Guangxi is located in southern China (20°54′–26°24′ N, 104°26′–112°04′ E) with a total area of 236,700 km^2^ [34] at the southeastern edge of the Yungui Plateau. The Tropic of Cancer crosses the central part of the region, and it borders Beibu Gulf in the south, the Nanling Mountains in the north, and extends to the Yungui Plateau in the west. It also belongs the transition zone from the Yungui Plateau to the southeastern coastal hills (Figure 1).

The topography of Guangxi is high in the northwest and low in the southeast, showing a northwest to southeast slope, mixed basin sizes, isolated mountain systems, intricate hills, and a wide distribution of stone desertification and a unique karst landscape [35]. Guangxi is located at low latitudes and has both a central subtropical monsoon climate and southern subtropical monsoon climate, with an average annual temperature above 16 °C, an average annual rainfall above 1100 mm, high temperatures and precipitation in summer, and short sunshine hours and dry and warm weather in winter [36]. Guangxi has many rivers and abundant hydraulic resources, and Xijiang River is the largest river in the region. Guangxi ranks third in China in terms of biodiversity and has an important position in the strategic pattern of national ecological security and ecological civilization construction. By the end of 2020, the total regional population of Guangxi was 57.18 million and its economy was relatively backward, with a gross domestic product (GDP) of CNY 2.22 trillion and a per capita regional GDP of CNY 44,309.

### 2.2. Data Source and Pre-Processing

The data required for this study include normalized difference vegetation index (NDVI) data, precipitation data, digital elevation model (DEM) data, land use data, vegetation type data, net primary productivity (NPP) data, soil data, fractional vegetation cover (FVC) data, road data, and evapotranspiration data. The data sources and related information are shown in Table 1. In order to ensure that the data resolution and projection information are consistent, the above data were reprojected and resampled by ArcGIS so that the spatial resolution was unified to 250 m, and the projection coordinate system was unified to the Albers projection.

### 2.3. Research Methodology

In this study, with reference to the “Guidelines for Evaluation of the Carrying Capacity of Resources and Environment and Suitability of Land Development (for Trial Implementation)” issued by the Ministry of Natural Resources of China in June 2019 and January 2020 and related research, combined with the landscape characteristics and ecological environment of Guangxi, three evaluation indexes of water conservation function, soil and water conservation function, and biodiversity maintenance function were selected to evaluate the importance of the ecosystem service function of Guangxi. In addition, two evaluation indexes of soil erosion sensitivity and rocky desertification sensitivity were selected to evaluate the ecological environment sensitivity of Guangxi; the importance of the ecological protection of Guangxi was obtained through superposition analysis. Based on the evaluation results of the importance of ecological protection in Guangxi, suitable patches were selected as ecological sources, while seven indicators such as elevation, slope, NDVI, land use, ecological risk, and main roads (railroads and highways) were selected to construct ecological resistance surfaces from both natural conditions and socio-economic perspectives. Additionally, ecological corridors, ecological pinch points, and ecological barrier points were identified using circuit theory. Combined with the research results, we put forward implementable suggestions for ecological construction and ecological protection in Guangxi. The specific research methods and contents of this study are shown in the flow chart (Figure 2).

#### 2.3.1. Evaluation of the Importance of Ecosystem Service Functions

Ecosystem service functions refer to the provisioning, regulating, cultural, and supporting services that ecosystems provide to humans and are usually assessed using model assessment methods [37] and quantitative indicators of net primary productivity (NPP) [38]. This study adopts the model assessment method to assess the water conservation function and soil and water conservation function and uses the quantitative index method of NPP to assess the biodiversity maintenance function.

(1)Water conservation function

Water conservation is the interaction of an ecosystem with water through its unique structure, thereby improving hydrology conditions and regulating the regional water cycles.

At present, the methods for evaluating the water conservation function are the precipitation saving method [39], forest canopy retention method [40], and the water balance method [41]. This study adopts the water balance equation to calculate the total water conservation in Guangxi. The calculation formula is as follows:(1)$$TQ=\sum_{i=1}^{j}\left(P_{i}-R_{i}-ET_{i}\right)\times A_{i}\times 10^{3}$$
(2)$$R_{i}=P_{i}\times \alpha$$
where *TQ* is the total water conservation; *P_i_* is the precipitation; *R_i_* is the surface runoff, ET_i_ is the evapotranspiration, *A_i_* is the area of type *i* ecosystem in the study area, *j* is the number of ecosystem types in the study area, and *α* is the average surface runoff coefficient.

(2)Soil and water conservation function

Soil and water conservation is an ecosystem through its structure and process that strives to reduce soil erosion caused by water erosion and is a guarantee of sustainable ecological, economic, and social development in mountainous areas.

The current methods for evaluating soil and water conservation function are the Revised Universal Soil Loss Equation (RUSLE) [42] and the NPP quantitative index assessment method [43]. In this study, the RUSLE model method was selected to calculate the amount of soil and water conservation in Guangxi. The calculation formula is as follows:(3)$$A_{c}=R\times K\times L\times S\times \left(1-C\right)$$
where *A_c_* is the amount of soil conservation, *R* is the rainfall erosion factor, *K* is the soil erodibility factor, *L* is the slope length factor, *S* is the slope factor, and *C* is the vegetation cover factor.

(3)Biodiversity maintenance function

Biodiversity is the ecological complex formed by organisms and their environment, and the biodiversity maintenance function refers to the role played by the ecosystem in maintaining species and gene diversity. The study of Guangxi’s biodiversity maintenance function is conducive to the formulation of biodiversity conservation programs and the promotion of a virtuous cycle of ecosystems. The current methods for evaluating biodiversity maintenance functions include meta-analysis [44], species distribution models [45], and the InVEST model method [46]. In this study, according to the data collection conditions and the actual situation in Guangxi, the NPP method was selected to calculate the biodiversity maintenance function. The calculation formula is as follows:(4)$$S_{bio}=NPP_{mean}\times F_{pre}\times F_{tem}\times \left(1-F_{alt}\right)$$
where *S_bio_* is the biodiversity maintenance function index, *NPP_mean_* is the multi-year NPP mean, *F_pre_* is the multi-year mean precipitation parameter, *F_tem_* is the multi-year mean temperature parameter, and *F_alt_* is the elevation parameter, and each parameter was subject to normalization.

#### 2.3.2. Ecological Environment Sensitivity Assessment

Ecological sensitivity is the sensitive response and self-recovery ability of an ecosystem to external disturbances at a specific temporal and spatial scale. Guangxi has a unique karst landscape with high mountains and steep slopes, and rocky desertification is widely distributed and soil erosion is serious. Therefore, two evaluation indexes of soil erosion sensitivity and rocky desertification sensitivity were selected in this study to evaluate the sensitivity of Guangxi’s ecological environment.

(1)Soil erosion sensitivity

In this study, the factors of rainfall erosion force, soil erodibility, terrain relief, and vegetation cover were used to evaluate the sensitivity of soil erosion. The above evaluation factors were divided into five sensitivity levels according to research needs: high sensitivity, medium–high sensitivity, medium sensitivity, low–medium sensitivity, and low sensitivity, and the graded values were assigned as 9, 7, 5, 3, and 1, respectively. A higher value indicates a higher sensitivity level, and the erosion sensitivity value was calculated according to the following formula:(5)$$SS_{i}=\sqrt[{4}]{R_{i}\times K_{i}\times LS_{i}\times C_{i}}$$
where *SS_i_* is the soil erosion sensitivity index, *R_i_* is the rainfall erosion force factor, *K_i_* is the soil erodibility erosion factor, *LS_i_* is the terrain relief factor, and *C_i_* is the vegetation cover factor. The assignment of each factor is shown in the table below (Table 2):

(2)Rocky desertification sensitivity

In this study, we used ecosystem type, topographic slope, and vegetation cover as the factors to carry out the sensitivity study of rocky desertification and divided the above evaluation factors into five sensitivity levels according to research needs: high sensitivity, medium–high sensitivity, medium sensitivity, low–medium sensitivity, and low sensitivity and assigned the values of 9, 7, 5, 3, and 1, respectively. A larger value indicates a higher sensitivity, and the sensitivity value of stone desertification in Guangxi was calculated according to the following formula (Table 3):(6)$$S_{i}=\sqrt[{3}]{D_{i}\times P_{i}\times C_{i}}$$
where *S_i_* is the rocky desertification sensitivity index, *D_i_* is the ecosystem type, *P_i_* is the topographic slope, and *C_i_* is the vegetation cover. The assigned values of each factor are shown in the following table:

#### 2.3.3. Evaluation of the Importance of Ecological Protection

The three functions of water conservation, soil and water conservation, and biodiversity maintenance were normalized separately, and 30% and 70% of the total value of the cumulative service functions of the type of ecosystem services were selected as the threshold values, which were classified into three levels: extremely important, important, and generally important. At the same time, the soil erosion sensitivity index and rocky desertification sensitivity index were classified into five levels of low sensitivity, low–medium sensitivity, medium sensitivity, medium–high sensitivity, and high sensitivity according to the natural breakpoint method.

The importance of the ecosystem service function was obtained by superimposing the results of the water conservation function, soil and water conservation function, and the biodiversity maintenance function. The calculation formula is as follows:(7)$$ESC=Max\left\{ES_{w},ES_{s},ES_{b}\right\}$$
where *ESC* is the evaluation result of the importance of the ecosystem service function, *ES_w_* is the evaluation result of the importance of the water conservation function, *ES_s_* is the evaluation result of the importance of the soil and water conservation function, and *ES_b_* is the evaluation result of the importance of the biodiversity maintenance function. The ecological sensitivity evaluation method is the same. In addition, according to the above formula, the results of the ecosystem service function importance evaluation and ecological environment sensitivity evaluation were superimposed to obtain the ecological protection importance evaluation results [47].

#### 2.3.4. Ecological Source Identification and Resistance Surface Construction

Ecological sources are important patches for maintaining regional ecological security [48] and are the basis for the construction of ecological security patterns whose identification is mainly based on ecosystem service functions [49], habitat quality [50], ecological red lines [51], etc. In this study, the extremely important area of the ecosystem service function and the extremely sensitive area of the ecological environment were spatially integrated. In order to facilitate corridor simulation, patches of 100 km^2^ or more were selected as ecological sources with reference to previous related studies [52,53].

The resistance surface refers to the resistance that a species receives when migrating. In this study, seven indicators such as elevation, slope, NDVI, land use, ecological risk, and major roads (railroads and highways) were selected to construct resistance surfaces from both natural conditions and socio-economic perspectives. Referring to the research results of previous scholars [54,55], a drag coefficient of 1–5 was assigned, and the higher the drag coefficient, the higher the value of drag to which the species is subjected during migration (Table 4). This study used hierarchical analysis to determine the weights of each index, and the consistency test result (CR = 0.0116, which is <0.1) indicated that the weight assignment met the requirements.

#### 2.3.5. Ecological Corridor Construction and Identification of Pinch Points and Barrier Points

Ecological corridors are channels for the flow of materials and energy between ecological sources [56]. This study evaluates the least-cost path based on the connection model and the random wandering model in circuit theory. Ecological corridors were obtained using Linkage Mapper, the linkage pathways tool module of GIS. Previous studies have shown that the smaller the cost-weighted distance to the minimum cost path ratio is, the stronger the connectivity of ecological corridors [57]. Based on this, this study classifies ecological corridors into three categories according to their connectivity size: general corridors, important corridors, and key corridors, where key corridors have the strongest connectivity.

The pinch point is a high-density area of current in the ecological network with important connectivity functions and a high potential for species migration through this area, making it a key location for preventing habitat degradation [58]. In this study, the cumulative current values in the study area were obtained through the Pinchpoint Mapper module in “all to one” mode, and the cumulative currents were classified into five levels according to the natural breakpoint method: level 1, level 2, level 3, level 4, and level 5, where the maximum range of cumulative current values in level 5 was the pinch point.

The barrier point is the area where species migration is more impeded, and restoration would significantly enhance connectivity between ecological sources [59]. In this study, based on the Barrier Mapper module with a moving window, 300 m was set as the minimum search radius, 3000 m was set as the maximum search radius, and 200 m was the step size. After comparison study, 900 m was selected as the best search radius to obtain the cumulative current recovery value. According to the natural breakpoint method, the cumulative current recovery value was divided into five levels (level 1, level 2, level 3, level 4, and level 5), where the five levels of cumulative current recovery value are the largest range of values for the barrier point.

#### 2.3.6. Research Limits

Based on the actual situation and data availability in Guangxi, this study adopts the model assessment method to assess the water conservation function and soil and water conservation function and uses the quantitative index method of NPP to assess the biodiversity maintenance function. Among them, the model assessment method is relatively accurate, but requires multiple parameters, has a large data demand, and requires high data quality. The NPP quantitative assessment method is simple to operate and has relatively few parameters, but the scope of application is influenced by locality, has a high degree of uncertainty and one-sidedness, and is suitable when data access is limited.

## 3. Results and Analysis

### 3.1. Evaluation of the Importance of Ecological Protection

#### 3.1.1. Evaluation of the Importance of Ecosystem Service Functions

In general, the ecosystem service function in Guangxi is categorized by important grade and very important grade (Figure 3d), occupying 83.11% of the total area of Guangxi (Table 5). The area with the important grade of ecosystem service function spans 122,946.00 km^2^, mainly scattered in the northwest, central, and southern areas of Guangxi and accounts for 52.35% of the total area of Guangxi. The area with the very important grade of ecosystem service function is the second largest at 72,231.75 km^2^ and accounts for 30.76% of the total area of Guangxi. It is mainly concentrated in Hechi, Liuzhou, and Guilin in northern Guangxi.

The ecosystem service function of Guangxi is dominated by the biodiversity maintenance function and water conservation function. Among them, the important area of the biodiversity maintenance function is 92,857.25 km^2^, mainly scattered in central Guangxi (Figure 3c); the extremely important area of the biodiversity maintenance function is 23,408.94 km^2^, accounting for 10.36% of the total area of Guangxi, mainly concentrated in Hezhou and Wuzhou in eastern Guangxi. The importance level of the water conservation function gradually increases from south to north (Figure 3a), and the area of the generally important area and important area is more balanced; the generally important area is mainly distributed in the south area of Guangxi, the important area is mainly distributed in the central area of Guangxi, and the extremely important area is mainly distributed in Liuzhou and Guilin in the north of Guangxi, in which the extremely important area of water conservation spans 36,541.31 km^2^. The generally important area of soil and water conservation function in Guangxi is the largest (Figure 3b), encompassing 67.45% of the total area of Guangxi and is mainly concentrated in the south of Guangxi. The distribution of the important area and the extremely important area is more fragmented.

#### 3.1.2. Ecological Sensitivity Evaluation

The overall ecological sensitivity of Guangxi is not high (Figure 4c), and the medium–highly sensitive and highly sensitive areas account for a total of 28.64% of the total area of Guangxi (Table 6), mainly concentrated in Hechi, Liuzhou, and Guilin in northern Guangxi. Among them, highly sensitive areas comprise the smallest proportion with an area of 20,911.00 km^2^, accounting for only 8.85% of the total area of Guangxi, and are scattered in the northern part of Guangxi.

Soil erosion sensitivity plays a dominant role in the ecological environment sensitivity of Guangxi (Figure 4a), and the low sensitivity, low–medium sensitivity, and medium sensitivity areas of soil erosion are all relatively balanced proportions at about 25.00%, while the proportion of highly sensitive areas is the least, occupying only 8.03% of the total area of Guangxi, and is mainly distributed sporadically in the northern part of Guangxi. From a spatial distribution perspective, the sensitivity level of soil erosion in Guangxi decreases gradually from north to south. The sensitivity of rocky desertification in Guangxi is weak (Figure 4b), and the highly sensitive area is the least proportion, occupying only 1.50% of the total area of Guangxi; the spatial distribution of each grade is fragmented.

The main importance levels of ecological protection in Guangxi are the important and extremely important levels (Figure 5). The area of ecological protection of the important area is 131,553.63 km^2^, accounting for 55.69% of the total area of the study area, mainly concentrated in the western and central areas of Guangxi. The area of ecological protection of the extremely important area is 83,802.25 km^2^, accounting for 35.47% of the total area of Guangxi, mainly concentrated in Hechi, Liuzhou, and Guilin in northern Guangxi, and Hezhou and Wuzhou in eastern Guangxi.

### 3.2. Building an Ecological Security Pattern

#### 3.2.1. Resistance Surface Construction

High resistance areas of elevation and slope in Guangxi are mainly concentrated in the northwestern and northeastern areas of Guangxi (Figure 6), high resistance areas of NDVI and land use are mainly scattered in the central and southern areas of Guangxi, and high resistance areas of ecological risk are obviously distributed and concentrated in the central and southern areas of Guangxi. The integrated resistance value in Guangxi is between 1.000 and 4.6291; the integrated high resistance areas are mainly distributed in central Guangxi (Laibin, Nanning, and Guigang) and southern Guangxi (Beihai); and the integrated low resistance areas are mainly distributed on the edge of Guangxi.

#### 3.2.2. Ecological Sources and Ecological Corridors

The total area of ecological sources in Guangxi is 60,556.99 km^2^, with 50 patches concentrated in the north and northeast of Guangxi (Figure 7). From the distribution of land use types, the coverage rate of forest land in ecological source land reaches 78.08%, and the proportion of unused land is very small (Table 7).

There are 115 ecological corridors in Guangxi with a total length of 4004.52 km (Table 8). Among them, there are 41 key corridors with a total length of 1132.60 km and 14 with a length of more than 10 km. The longest key corridor runs from the east of Wuzhou to the west of Yulin with a length of 194.97 km. There are 40 important corridors with a total length of 1407.62 km, and the longest one reaches 166.74 km, spanning from the southernmost part of Hechi longitudinally through Baise and Nanning to the eastern part of Chongzuo. The number of general corridors is the least, but the overall length is the largest. In terms of spatial distribution, the key corridors are mostly concentrated longitudinally in the western and eastern fringes of Guangxi, the important corridors are scattered around key corridors, and general corridors are mostly in central Guangxi in Laibin, Nanning, Guigang, and Qinzhou cities. On the whole, it shows that the importance of ecological corridors gradually decreases from the exterior to the middle.

#### 3.2.3. Cumulative Current Value and Cumulative Current Recovery Value

The cumulative current value in Guangxi is between 0~0.1279, and the total area is 74,478.59 km^2^, occupying a total of 31.35% of the total area of Guangxi (Figure 8a). The cumulative currents are mainly dominated by secondary, tertiary, and primary cumulative current values (Figure 9), with areas of 33,373.31 km^2^, 18,590.03 km^2^, and 17,168.88 km^2^, respectively, occupying a total of 92.82% of the total cumulative current area. The five-level cumulative currents, that is, the area of the pinch point, only occupies 0.90% of the total area of cumulative currents.

The cumulative current recovery value in Guangxi is between 0 and 2.4482, with a total area of 24,662.50 km^2^, occupying a total of 10.42% of the total area of Guangxi (Figure 8b). The cumulative current recovery values are mainly secondary, primary, and tertiary, with areas of 8230.31 km^2^, 7921.19 km^2^, and 5983.19 km^2^, respectively, occupying a total of 79.45% of the total cumulative current recovery area. The five-level cumulative current recovery value, that is, the barrier point, only occupies 6.74% of the cumulative current recovery value. In terms of spatial distribution, the level of the cumulative current recovery value gradually decreases from the surrounding area to the middle, in which the fourth- and fifth-level cumulative current recovery value is mainly distributed in the central area of Guangxi.

#### 3.2.4. Pinch Points and Barrier Points

There are 301 pinch points in Guangxi (Figure 10), with an area of 669.44 km^2^. The spatial distribution is more fragmented and is mainly distributed on general corridors and near ecological sources. From a land use type viewpoint, the pinch points are mainly distributed on forest land, accounting for 68.42% of the total area of the pinch points, followed by unused land and arable land, and the least amount of pinch points are on construction land.

There are 286 barrier points in Guangxi, with an area of 1878.75 km^2^, concentrated on important corridors and general corridors in central Guangxi. From a land use type perspective, barrier points are mainly distributed on arable land, accounting for 74.76%, followed by forest land, and the least amount of barrier points are found on unused land.

## 4. Discussion

### 4.1. Analysis of Ecological Protection Importance Evaluation

Guangxi is rich in water resources and biodiversity, high mountains and steep slopes, extensive rock desertification, and serious soil erosion. There are obvious regional differences in the evaluation results of ecological protection importance, which are closely related to the natural factors and human activities in each region. Among them, the natural factors are mainly: topography, geology, precipitation, and other factors. The main characteristics of Guangxi topography are more mountains and less plains and high mountains and steep slopes, while the slopes are extremely prone to soil erosion. The soil in this area is mostly formed by rock weathering, and this soil has loose structure and poor water storage capacity, and high temperature and rain are very likely to cause collapse of steep slope sections. Precipitation is an important factor causing regional soil erosion and rock desertification, and is also an essential influencing factor for water connotation. The rainfall in Guangxi is high in the north and low in the south, and the rainfall is concentrated with high intensity, which is easy to cause landslides and soil erosion, so the water connotation function, soil erosion, and stone desertification sensitivity in northern Guangxi is obviously greater than that in southern Guangxi. A forest is an important ecosystem for water conservation and an essential habitat for maintaining biodiversity. Woodlands in Guangxi are widely distributed with an area of 154,957.78 km^2^, mainly concentrated in the southern part of Hechi, Hezhou, and Wuzhou, which is the reason why the function of water connotation and biodiversity maintenance in this region has become a very important area.

With the acceleration of economic construction and urbanization, the contradiction of more people and less land has increased human demand for land, especially in the central region of Guangxi where human activities are intense and construction land continues to expand, which affects the evaluation results of ecological protection importance in the region. Secondly, indiscriminate logging, excessive mining, and mining have contributed to the continuous reduction in forest coverage and serious soil exposure. In addition, there is a large number of steep slopes to open up wasteland in Guangxi, cutting down the original vegetation to plant fruit trees and crops with economic value, resulting in the intensification of soil erosion and rock desertification.

The reduction in water conservation and soil and water conservation capacity, loss of biodiversity, soil erosion and increased rock desertification will inevitably bring a series of problems, such as land degradation and serious landscape fragmentation, which in turn restricts the development of local agriculture and tourism. Secondly, soil erosion leads to increased rock desertification and the formation of stone desertification in the process of causing more serious soil erosion; this vicious circle will lead to frequent occurrence of natural disasters such as droughts and landslides, affecting industrial and agricultural production and the safety of people’s lives and property.

Therefore, it is urgent to carry out ecological security protection and improvement work reasonably. It is necessary to regularly carry out ecological security assessment work, establish and improve the management mechanism and policies and regulations for ecological security protection, and carry out ecological restoration work in different regions, such as: vegetation restoration, returning farmland to forests, industrial poverty alleviation, and other measures.

### 4.2. Analysis of Ecological Security Patterns

In 2012, the Guangxi Zhuang Autonomous Region People’s Government issued the “Guangxi main functional area plan”, which pointed out that the ecological security pattern of Guangxi is “two screens, four areas and one corridor”. The two screens refer to the western Guangxi ecological barrier and the coastal ecological screen of Beibuwan; the four areas refer to the northeast ecological functional area of Guangxi, the southwest ecological functional area of Guangxi, the central ecological functional area of Guangxi, and the Shiwan mountain ecological reserve; the one corridor refers to the Xijiang Qianli Green Corridor. The spatial distribution and main functions of the four functional areas in the plan are basically consistent with the results of this study, but there are differences in the ecological screens and ecological corridors, among which the Beibuwan coastal ecological screen is an ecological screen constructed mainly by coastal windbreak forest and marine ecological restoration, while this study only focuses on ecological construction with Guangxi land, and further marine ecological protection is needed in the future. For the identification of ecological corridors in Guangxi, the planning targets afforestation and water ecological environment protection and the area along the Xijiang River as the ecological corridor in Guangxi, but this study identifies multiple ecological corridors by combining the resistance surface and the least-cost path.

This study shows that the ecological security pattern of Guangxi consists of ecological sources, ecological corridor, ecological pinch point, and ecological barrier point. Ecological sources are the key area for regional ecological construction and ecological protection, and targeted protection and construction should be carried out in combination with the main functions of ecological sources. There are 115 ecological corridors in Guangxi, ecological sources are connected with each other by ecological corridors, and there are 63 corridors with lengths over 10 km. Excessive lengths will make ecological corridors more sensitive and reduce their resistance to internal and external disturbances. Therefore, it is necessary to focus on protecting the environment around key ecological corridors, while enhancing the number and area of ecological sources, thus reducing the length of ecological corridors and strengthening the stability and circulation of ecological corridors. Guangxi ecological pinch points and barrier points are mostly distributed on ecological corridors, and the ecological resistance around these areas should be appropriately reduced to enhance landscape connectivity and improve the stability and anti-disturbance ability of ecological security patterns.

### 4.3. Study Shortcomings

Due to the limitations of data collection and model accuracy, this study only combined the results of ecosystem service function importance and ecological sensitivity evaluation to identify ecological sources, without considering local nature reserves, scenic spots, and restricted development zones, etc. Therefore, future research needs to consider many aspects when identifying ecological sources, and needs to combine nature reserves, scenic spots, and restricted development areas zones, etc. Second, the methods and standards for the construction of ecological resistance surfaces are not yet unified, resulting in differences in ecological security patterns, hence it is necessary to formulate targeted ecological resistance surface construction methods and standards in combination with the characteristics of the regional ecological environment. In addition, the research on the optimization and management of ecological safety patterns is relatively weak, the current ecological pattern research mainly focuses on the construction of ecological security patterns, and there is less research on subsequent optimization and management, which needs to consider the optimization and management of ecological security patterns in future research.

## 5. Conclusions

This study identifies ecological sources based on the importance of ecological protection, selects resistance factors from both natural factors and human activities to construct a resistance surface, and uses circuit theory to construct an ecological security pattern. The conclusions are as follows:(1)The very important area of ecosystem service function in Guangxi accounts for 30.76% of the total area of Guangxi, mainly concentrated in the northern and northeastern areas of Guangxi; the highly sensitive area of ecological environment in Guangxi accounts for 8.85% of the total area of Guangxi, scattered in the northern part of Guangxi.(2)The total area of ecological sources in Guangxi is 60,556.99 km^2^, with 50 patches, concentrated in the northern and northeastern areas of Guangxi, mainly in woodlands.(3)There are 115 ecological corridors in Guangxi with a total length of 4004.52 km, among which there are 41 key corridors with the largest number but the smallest length, and the longest corridor is 194.97 km. The spatial distribution of ecological corridors shows a trend of gradually decreasing the importance from the surrounding to the middle ecological corridors.(4)There are 301 ecological pinch points in Guangxi, with an area of 669.44 km^2^, and the spatial distribution is relatively fragmented and mainly distributed on the forest land. There are 286 barrier points in Guangxi with a total area of 1878.75 km^2^, mostly distributed on important corridors and general corridors in central Guangxi.

Our research results show that we need to carry out targeted protection in conjunction with the main functions of ecological sources in Guangxi, increase the area and number of ecological sources, focus on protecting the environment around key ecological corridors, reduce the resistance around ecological pinch points and ecological barrier points, and improve the stability of ecological security patterns. In addition, our study provides referenceable opinions for developing countries and karst areas to construct ecological security patterns.

## Figures and Tables

**Figure 1 ijerph-19-05699-f001:**
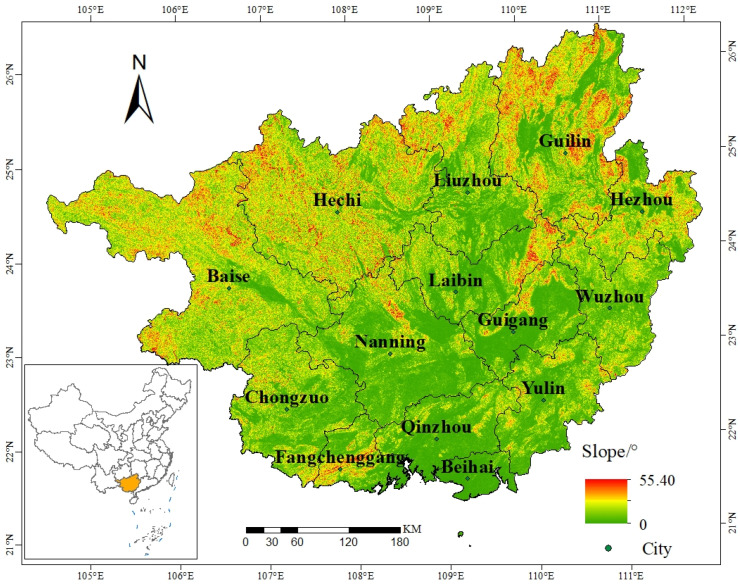
Location of the study area.

**Figure 2 ijerph-19-05699-f002:**
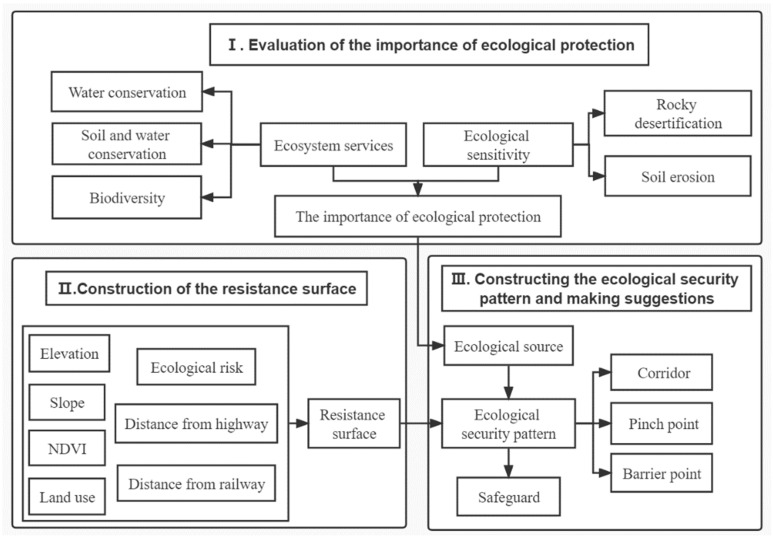
The framework of the research.

**Figure 3 ijerph-19-05699-f003:**
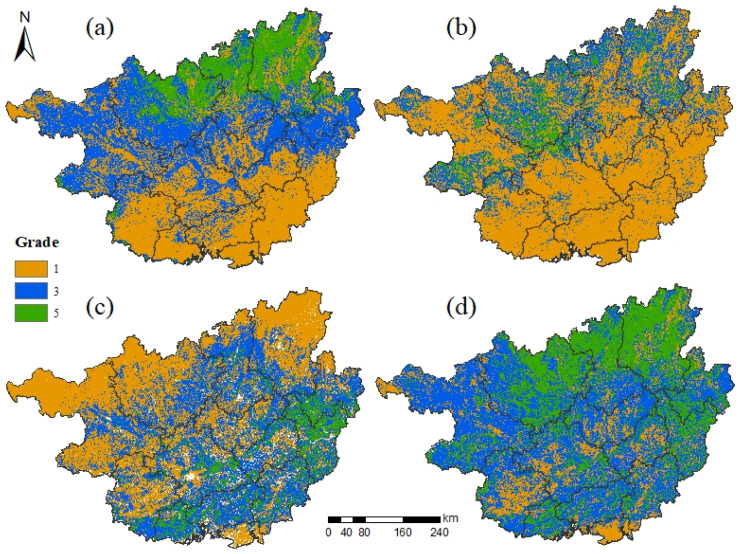
Evaluation of the importance of ecosystem service functions. Note: (**a**–**d**) indicate water conservation function, soil and water conservation function, biodiversity maintenance function, and ecosystem service function, respectively; 1, 3, 5 indicate generally important, important, and extremely important, respectively.

**Figure 4 ijerph-19-05699-f004:**
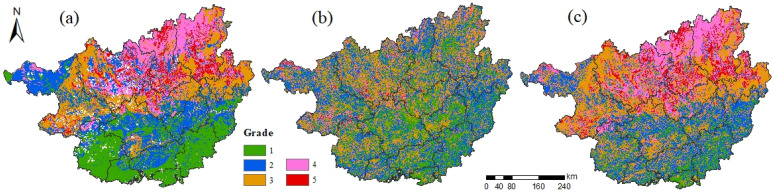
Ecological sensitivity evaluation. Note: (**a**–**c**) indicate soil erosion sensitivity, rock desertification sensitivity, ecological sensitivity, respectively; 1, 2, 3, 4, 5 indicate low sensitivity, low–medium sensitivity, medium sensitivity, medium–high sensitivity, high sensitivity, respectively.

**Figure 5 ijerph-19-05699-f005:**
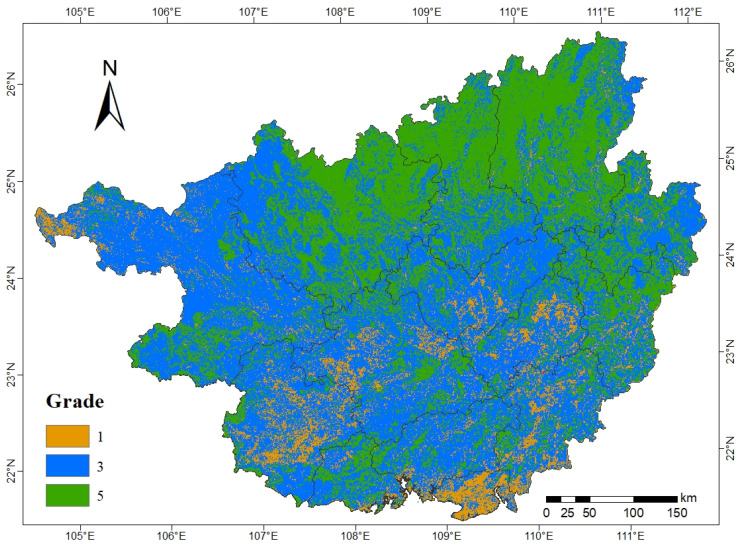
Ecological protection importance evaluation.

**Figure 6 ijerph-19-05699-f006:**
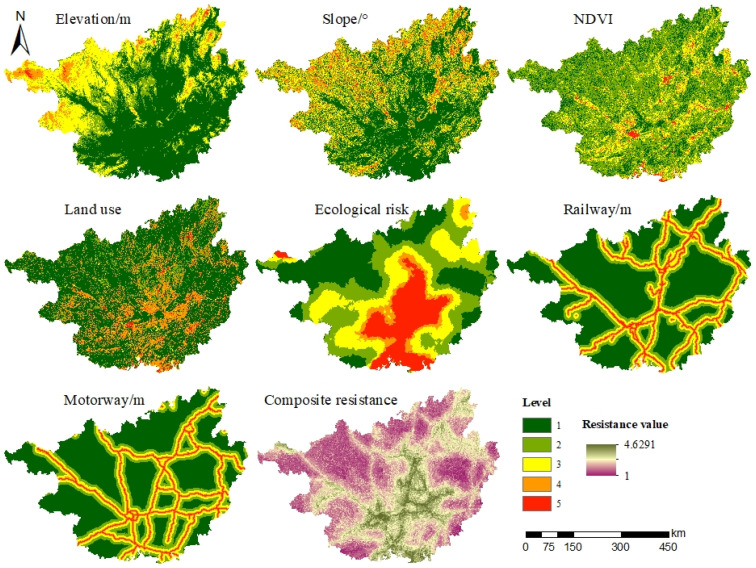
Spatial distribution of resistance factor and integrated resistance surface.

**Figure 7 ijerph-19-05699-f007:**
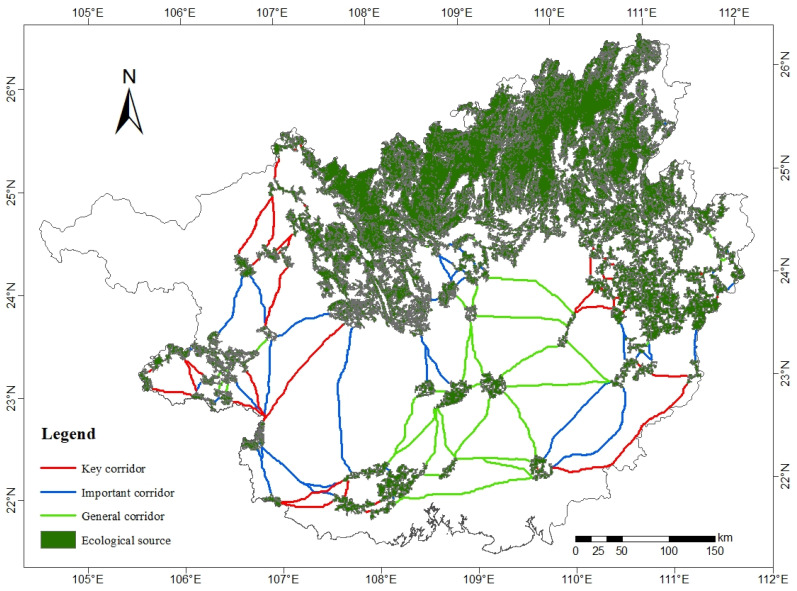
Spatial distribution of ecological sources and ecological corridors.

**Figure 8 ijerph-19-05699-f008:**
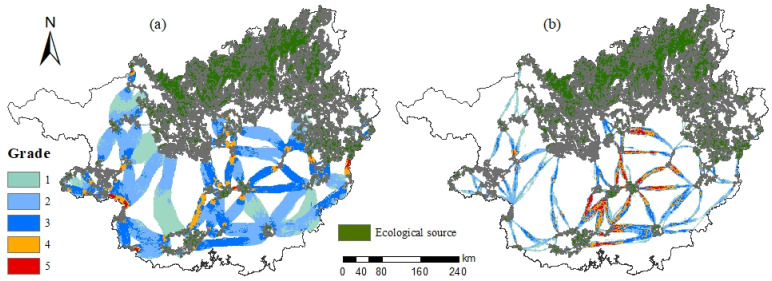
Spatial distribution of cumulative current value and cumulative current recovery value. Note: (**a**,**b**) denote cumulative current value reclassification, cumulative current recovery value reclassification, respectively.

**Figure 9 ijerph-19-05699-f009:**
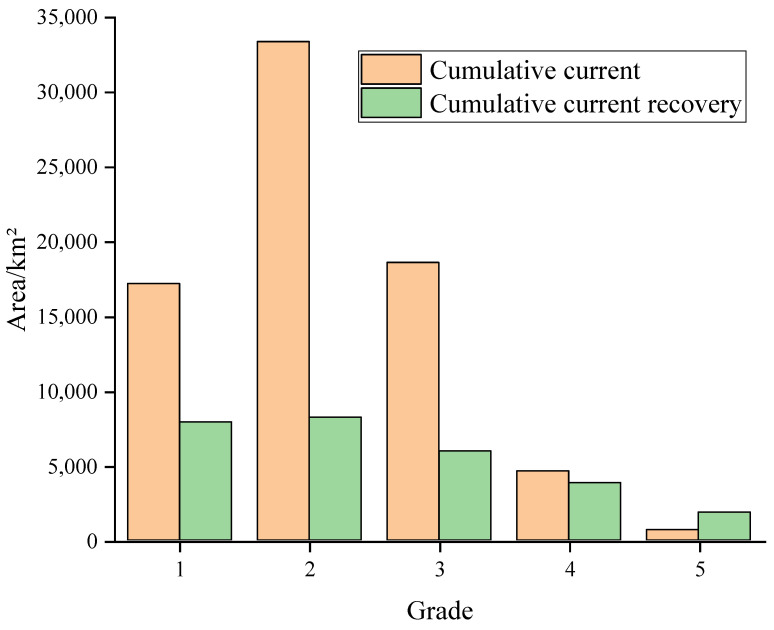
Area of each grade of cumulative current value and cumulative current recovery value.

**Figure 10 ijerph-19-05699-f010:**
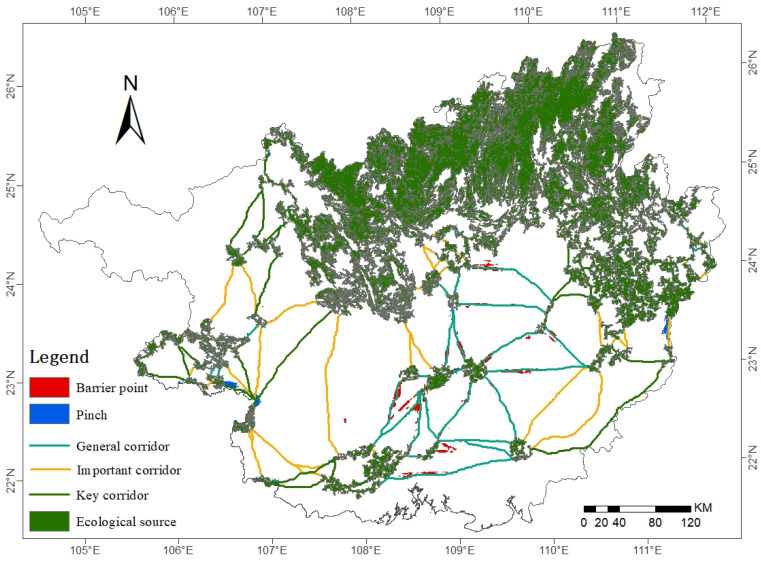
Ecological Security Patterns.

**Table 1 ijerph-19-05699-t001:** Data sources and descriptions.

Data Name	Data Source	Resolution	Data Description
Vegetation product data MOD13Q1	NASA (https://ladsweb.modaps. eosdis. nasa.gov/search/ (accessed on 20 December 2020))	250 m 16 d	Average NDVI from 2000 to 2018
Precipitation data	European Centre for Medium-Range Weather Forecasts (ECMWF) (https://www.ecmwf.int/ (accessed on 16 December 2020))	0.5°	Average precipitation from 2000–2018
DEM data	Geospatial Data Cloud (http://www.gscloud.cn/ (accessed on 7 March 2021))	30 m	Selection of elevation and slope
Land use data	Resource and Environment Science and Data Center of the Chinese Academy of Sciences (https://www.resdc.cn/ (accessed on 14 January 2021))	30 m	Land use data for 2018, with land use types of arable land, woodland, grassland, water, construction land, and unused land
Vegetation type data		1:1,000,000	The vegetation types are forest, scrub, swamp, farmland, grass, other
NPP data		1 km	NPP data from 2000–2010
Soil data	National Tibetan Plateau Data Center	1:1,000,000	China soil map based harmonized world soil database (HWSD) (v1.1) (2009)
FVC data	Beijing Normal University (http://glass-product.bnu.edu.cn/ (accessed on 27 December 2021))	500 m 8 d	Average FVC from 2000 to 2018
Road data	National Catalogue Service For Geographic Information (https://www.webmap.cn/ (accessed on 10 December 2021))	1:250,000	Select rail and highway
Evapotranspiration (ET) data	Figshare (https://figshare.com/ (accessed on 6 November 2021))	1 km	Average evapotranspiration from 2000 to 2018

**Table 2 ijerph-19-05699-t002:** Assignment of each factor of soil erosion sensitivity [47].

Evaluation Factors	High Sensitivity	Medium–High Sensitivity	Medium Sensitivity	Low–Medium Sensitivity	Low Sensitivity
Rainfall erosion force *R*	Graded assignment according to the natural breakpoint method				
Soil erodibility erosion *K*	Sandy, chalky soils	Sandy loam, powdered clay, loamy clay	Surface sandy soil, loamy soil	Coarse gravel, fine gravel, clay	Sand
Terrain relief LS	>300	100~300	50~100	20~50	0~20
Vegetation cover *C*	≤0.2	0.2~0.4	0.4~0.6	0.6~0.8	>0.8
Hierarchical assignment	9	7	5	3	1

**Table 3 ijerph-19-05699-t003:** Assignment of each evaluation factor of rock desertification sensitivity [47].

Evaluation Factors	High Sensitivity	Medium–High Sensitivity	Medium Sensitivity	Low–Medium Sensitivity	Low Sensitivity
Ecosystem type *D*	Bare ground, dryland, garden	Grassland	Scrub	Forest	Wetlands, construction land, water field
Topographic slope *P*	≥25°	15°~25°	8°~15°	5°~8°	≤5°
Vegetation cover *C*	≤0.2	0.2~0.4	0.4~0.6	0.6~0.8	>0.8
Hierarchical assignment	9	7	5	3	1

**Table 4 ijerph-19-05699-t004:** Graded assignment of resistance factors and weight assignment.

Resistance Factor	Tiered Metrics	Resistance Value	Weights
Elevation (m)	<300	1	0.0753
	300~500	2	
	500~1000	3	
	1000~1500	4	
	>1500	5	
Slope (°)	<5	1	0.1219
	5~10	2	
	10~15	3	
	15~25	4	
	>25	5	
NDVI	>0.9	1	0.0492
	0.8~0.7	2	
	0.7~0.8	3	
	0.6~0.7	4	
	>0.6	5	
Land use	Woodland	1	0.2057
	Grassland	2	
	Waters	3	
	Arable land	4	
	Construction land, unused land	5	
Ecological risk	Low risk	1	0.3422
	Low–medium risk	2	
	Medium risk	3	
	Medium–high risk	4	
	High risk	5	
Distance from railroads and highways (km)	>10	1	0.2057
	5~10	2	
	2~5	3	
	1~2	4	
	<1	5	

**Table 5 ijerph-19-05699-t005:** Area distribution and proportion of each ecological function importance level.

Grade	Water Connotation Function		Soil and Water Conservation Function		Biodiversity Maintenance Function		Ecosystem Service Function	
	Area (km^2^)	Area Ratio	Area (km^2^)	Area Ratio	Area (km^2^)	Area Ratio	Area (km^2^)	Area Ratio
Generally important	103,859.19	44.32	157,912.13	67.45	109,789.75	48.57	39,656.63	16.89
Important	93,961.88	40.09	56,691.88	24.21	92,857.25	41.08	122,946.00	52.35
Extremely important	36,541.318	15.59	19,527.50	8.34	23,408.94	10.36	72,231.75	30.76

**Table 6 ijerph-19-05699-t006:** Area distribution and proportion of each ecological sensitivity level.

Grade	Soil Erosion Sensitivity		Rock Desertification Sensitivity		Ecological Sensitivity	
	Area (km^2^)	Area Ratio	Area (km^2^)	Area Ratio	Area (km^2^)	Area Ratio
Low	53,875.94	23.73	58,077.56	24.60	21,702.00	9.19
Low–medium	61,367.81	27.03	85,562.69	36.24	62,982.13	26.67
Medium	54,617.88	24.06	73,500.19	31.13	83,844.06	35.50
Medium–high	38,941.06	17.15	15,460.00	6.55	46,730.94	19.79
High	18,230.44	8.03	3526.81	1.50	20,911.00	8.85

**Table 7 ijerph-19-05699-t007:** Area of each land use type in the ecological security pattern (unit: km^2^).

	Grassland	Arable Land	Construction Land	Woodland	Waters	Unused Land
Ecological Sources	6774.50	5052.50	560.38	46,795.13	750.50	1.50
Pinch point	46.13	94.06	11.758	537.00	10.50	85.38
Barrier point	113.63	1042.06	204.19	446.56	72.31	85.38

**Table 8 ijerph-19-05699-t008:** Summary of the number and length of ecological corridors.

Type	Number	Number Ratio	Length (km)	Length Ratio	Longest Corridor Length (km)
General corridor	34	29.57%	1464.30	36.57%	158.37
Important corridor	40	34.78%	1407.62	35.15%	166.74
Key corridor	41	35.65%	1132.60	28.28%	194.97
Total	115	100%	4004.52	100%	

## Data Availability

Not applicable.

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
