# Peer review of "Construction of Ecological Security Pattern Based on the Importance of Ecological Protection—A Case Study of Guangxi, a Karst Region in China"

_ijerph, 2022, doi:10.3390/ijerph19095699_

Round 1

Reviewer 2 Report

Methodologically, I think that "the vague part" of the study is represented by the input data, which are general and refferred to an extended scale. In order to frame this limit, I would suggest to better explain the contents of the "Section 4.3 Study Shortcomings" before, in a separate section (possibly after "Section 2.3 Research Methodology"), that explains, very well, the Research Limits. In this way, the reader is correctly conducted in the understanding of the applied method.

Lines 150-152: the source "Guidelines for Evaluation of the Carrying Capacity of Resources and Environment and Suitability of Land Development (for Trial Imple-mentation)" issued by the Ministry of Natural Resources should be defined (maybe the authors or the curators or the editor, year... any useful information).

Reviewer 3 Report

Please provide source information when presenting standard values in Table.

Few figures quality is scattered. Improve the resolution

What are the typical values for each of the contributing factors eg fig 6? It is not clear from the current figure.

In the ecological protection importance, why only 3 grades are reflected.

Overall the manuscript is well written and can be accepted for publication

Reviewer 4 Report

The manuscript addresses an interesting topic: ecological security in both developed and developing countries is a strategic value that involves not only the safeguarding of biotic resources, but also allows for the preservation of biodiversity in marginalised territories that are generally associated with fragile ecological conditions.

The introductory section manages a broad overview, explaining the issues surrounding the object of study, however, I have a concern, the authors handle key concepts that are not addressed and much less explained in the context of the study region, I recommend the authors to retake the section to make these theoretical improvements.  This section should also state precisely, at least at the beginning of the section, the objective of the research, which is not clear.

The methodological section is logically and coherently constructed, which makes the study robust and replicable. But this section should make it clear which approach was used in the research design, in this case the quantitative approach, as well as its scope and limitations.

The results of the study are interesting and highlight the importance of conducting this type of study, the ecological elements described in the study region show an intense use of the territory and the sensitivity to anthropogenic activities, I note that in the results precisely nothing is said about the human factor and its interactive dynamics.  The generation of patterns should not only be useful for the ecological characterisation of the region in a context of environmental vulnerability, but the contributions of the study should also include recommendations for public policies for the region and other regions of China with similar agro-ecological conditions.

The limitations of the study that are indicated at the end of the Discussion section should be part of the results section, and methodological strategies to overcome them should be raised in the discussion.

The conclusion is deficient and needs to be expanded, the results are abundant and of high quality and provide information that the authors do not highlight. It is necessary to provide a public policy proposal in the context of the agro-ecological conditions of the study area with an impact on policy recipients and policy makers. The authors should expand the bibliography.

Round 2

Reviewer 4 Report

The authors have improved the manuscript according to my recommendations, with the current presentation I have no objection to its publication.